# Effects of Changes in Acoustic and Non-Acoustic Factors on Public Health and Reactions: Follow-Up Surveys in the Vicinity of the Hanoi Noi Bai International Airport

**DOI:** 10.3390/ijerph17072597

**Published:** 2020-04-10

**Authors:** Thu Lan Nguyen, Bach Lien Trieu, Yasuhiro Hiraguri, Makoto Morinaga, Takashi Morihara, Takashi Yano

**Affiliations:** 1Department of Architectural Design, Interdisciplinary Faculty of Science and Engineering, Shimane University, 1060 Nishikawatsu-cho, Matsue, Shimane 690-8504, Japan; trieulien0903@gmail.com; 2Faculty of Architecture, Kindai University, 3-4-1 Kowakae, Higashiosaka City, Osaka 577-8502, Japan; hiraguri@arch.kindai.ac.jp; 3Defense Structure Improvement Foundation, 15-9 Yotsuya-Honshio-cho, Shinjuku-ku, Tokyo 160-0003, Japan; d4-morinaga@bsk-z.or.jp; 4Department of Architecture, National Institute of Technology, Ishikawa College, Tsubata, Kahoku-gun, Ishikawa 929-0932, Japan; morihara@ishikawa-nct.ac.jp; 5Kumamoto University, 2-39-1 Kurokami Chuo-ku, Kumamoto 860-8555, Japan; yano@gpo.kumamoto-u.ac.jp

**Keywords:** changed noise environment, aircraft noise, annoyance, health effects

## Abstract

Herein, the effects of changes in acoustic and non-acoustic factors on public health and reactions were assessed using two follow-up investigations; this was achieved after three surveys were conducted on the impact of the step change in noise caused by the increased number of flights at the Noi Bai International Airport in Hanoi (Vietnam) after the new terminal building was opened to the public. Exposure-response relationships established in the follow-up studies were less in number than those established in 2015 after the step change had occurred, and were almost similar to the relationship established in the survey conducted before the step change; however, these relationships were significantly greater than those established in the European Union position paper. Comparisons between respondents with high blood pressure and insomnia ratios at different noise level ranges showed that there is no significant association between ratios of high blood pressure and day-evening-night noise levels; however, an exposure-response relationship was discovered between insomnia and night-time noise levels. Non-acoustic factors such as noise sensitivity, sound insulation capacity of houses, and length of residence were found to curb the respondents’ annoyance, insomnia, and high blood pressure. Thus, an improvement in residence quality and a restriction on nighttime flight operation is necessitated.

## 1. Introduction

Increases in flight operations aimed at meeting growing air travel demand can have various negative impacts on the environment. In particular, this causes noise and air pollution, which affects the quality of life and health of communities living near the airport [1]. However, the number of studies on this issue in developing countries wherein the aviation transport business is showing the fastest growth rate is extremely limited [2]. The exposure-response relationships for noise annoyance were proposed based on the data from socio-acoustic surveys as a foundation for noise framework policy [3,4,5,6]. However, the majority of the surveys assumed a steady noise state whereby the amount of noise exposure did not change throughout the year. Furthermore, there are fewer studies involving the step change of noise exposure levels due to changes in airport operation conditions [7,8]. Recent investigations based on meta-analysis found that step change in traffic noise caused an “excess response” [9,10,11]. Brown and van Kamp defined the effects caused by noise exposure in steady-state conditions as the “exposure effect” and the additional effects caused by a change in noise exposure as the “change effect” [10]. An “excess response” is defined as the state whereby the response to an increase or decrease in noise exposure, results in a respective increase or decrease in the response as compared to the response in the steady state condition; the opposite is known as “under response.” In this paper, we used the aforementioned terms to describe the findings of the research. The development of air transport infrastructure, which is being actively promoted in developing countries, is facilitating the negative environmental changes in areas around the airports including noise problems. Therefore, understanding the impact of environmental change on people in order to appropriately manage aircraft noise is required; further, increasing the number of flights in accordance with the health and living quality of residents in the surrounding vicinities is essential.

A socio-acoustic survey on community response to aircraft noise around the Hanoi Noi Bai International Airport (NBIA) in Vietnam was conducted in 2009 [12]. The operation status of NBIA was considered to be stable around the survey period. Since then, the number of operations of the aircraft has gradually increased, especially after the opening of a new terminal building in December 2014. To assess the effects of a step change in noise exposure levels around NBIA, step-change surveys were conducted once before and twice after the change in operation. As a result, a significant change effect including the excess response and under response was observed with respect to annoyance; however, only the excess response was observed with regard to sleep disturbance [13]. The reaction to noise shortly and long after the step change in noise exposure may differ, as suggested in other studies [14,15]. As shown in the results of previous surveys in NBIA, the change effect was observed immediately after the step change occurred. However, whether the change effect would decline or remain at the same level could not be determined. To elucidate whether this change effect decreases over time or persists afterward, two follow-up surveys were conducted in 2017 and 2018, approximately 3 and 4 years, after the step change, respectively.

In tandem with the change in air transport, there has been a drastic change in Vietnam’s economy and urbanization in recent years. As a result, the housing conditions of the Vietnamese people in general, as well as of those living around NBIA, were observed to change. Noise annoyance was found to vary among factors other than noise exposure such as housing, neighborhood environment, socio-demographic variables, and personal as well as environmental contexts [16,17]. The effect of noise change should be investigated using the effects of both acoustic and non-acoustic variables. In this paper, the results of socio-acoustic surveys over five years, from 2014 to 2018 around NBIA will be summarized for the purpose of assessing the effects of changes in operational and residential factors on public health and reactions in the vicinity of NBIA. The outcomes of this study are expected to contribute to the establishment of appropriate noise policies for improving the living environment around the airports in developing countries.

## 2. Materials and Methods

### 2.1. Survey Sites

NBIA has two parallel runways in the east-west direction (11L–29R and 11R–29L). Since the operational direction of the runway is influenced by the wind direction, the use of the flight path toward the east occupied nearly 90% of the total movements at NBIA. As shown in Figure 1, in the surveys from 2014 to 2018, there were a total of 13 sites (Sites A1–A13); these included three sites (A5, A6, and A8) located close to the end of the runway of the airport, and two sites (A12 and A13) located in the northeast direction of the airport. The sites A12 and A13 have almost the same living environment as the other sites but aircraft noise was assumed to have an insignificant effect on them.

### 2.2. Socio-Acoustic and Health Surveys

In the series of surveys, Vietnamese questionnaires including two standardized annoyance questions recommended by ICBEN [18,19,20] were prepared. Community responses from the people living in the vicinity of NBIA were collected through the face-to-face interview method. The percentage of respondents who were highly annoyed (% HA) was considered as the percentage of respondents who chose 8, 9, or 10 out of the 11-point numerical scale (0–10). In the surveys since 2014, the percentage of insomnia (% ISM) was considered as the frequency of effects on sleep as proposed in previous studies [21,22,23] and was used as an indicator of the effect that flight operation during the nighttime had on sleep. Respondents with insomnia referred to those who responded affirmatively to “have any trouble with sleep” and “sleepy during daytime and cannot work well more than three times a week” and had experienced at least one of the other symptoms (1)–(4) listed in the Insomnia Symptom Questionnaire (ISQ) more than three times in a week. The wordings of the questions are listed in Table 1.

Sensitivity was recognized as a moderator that change the effect of environmental noise exposure on health outcomes. In the present study, noise sensitivity was included in the questionnaire of all the surveys, among one of the seven items enquiring about sensitivity by a question was termed “In daily life, climatic factors as well as environmental conditions affect us much, then how much are you sensitive to the following factors?” The respondents were asked to respond to each item on a five-point scale 1: Not at all; 2: Slightly; 3: Moderately; 4: Very; 5: Extremely. The question was placed after those inquiring annoyance and before the demographic questions.

In addition to general annoyance and impacts on sleep, exposure to high levels of aircraft noise may adversely affect people with cardiovascular disease and other ailments [1]. Since studies of health effects of aircraft noise have not been conducted for residents living near airports in developing countries, in the survey of 2017, data on the health status of residents such as body mass index (BMI) and blood pressure based on self-report were collected to evaluate the effects of aircraft noise on the health of the population around NBIA. A smaller number of respondents in the 2018 survey were randomly selected from the same residential areas as those of the respondents in the 2017 survey. Furthermore, in the survey in 2018, the blood pressure of all residents was measured using a blood pressure meter (HEM-6324T, OMRON, Kyoto, Japan). Instead of questions about living conditions and the surrounding environment, questions about current health statuses such as BMI, blood pressure and heart rate were added.

### 2.3. Noise Estimation

Predicted values for the estimation of noise exposure to respondents was preferable in all surveys. However, data required for prediction such as flight route, runway use, flight operation data, and airplane performance could not be obtained in the 1st, 2nd, and 3rd surveys. Therefore, field measurement values were used for the estimation of noise exposure instead of noise prediction in the first three surveys. Aircraft noise exposure was measured by using sound level meters (RION NL-42, NL-21, NL-22) with microphones covered with all-weather windscreens and positioned on the rooftops per site. The microphones were set at the height of 1.5 m above the roofs and at least 1 m away from any other reflecting surface. The houses selected for noise measurements were the highest ones in the areas and has the distance from the other houses in the areas of the questionnaire survey not exceed 500 m.

Day-evening-night-weighted sound pressure level (*L*_den_) and nighttime equivalent continuous sound pressure level (*L*_night_) were estimated from the field measurement of noise levels. A-weighted and S-weighted sound pressure levels (*L*_A,S_) sampled at 1 s were recorded continuously over 7 days. The noise data of each day for each site was compared with flight logs to identify the aircraft events and then calculate the *L*_den_. Since the day, evening, and night periods are different among countries, depending on the activity pattern of daily life, in this study, they are defined as the periods from 6:00 to 18:00, from 18:00 to 22:00, and from 22:00 to 6:00, respectively.

Regarding the validity of the one-week measurement, one-year average forecasts are desirable for assessment of noise exposure of the respondents, as social surveys typically assess annoyance over the past year. However, since the present study is about a change effect of noise, and the period of evaluation was “the last one month” instead of the generally used “last 12 months,” weekly average data was assumed to be sufficient to represent noise exposure for a month.

Since sufficient data for noise prediction was obtained in the surveys in 2017 and 2018, *L*_den_ and *L*_night_ were estimated from noise contour maps calculated using the Integrated Noise Model (INM) [24] instead of field measurements. The necessary data for calculating the noise contour maps such as airport operation data including flight logs and weather conditions during the surveys were provided by the airport managers. The flight operation at NBIA is categorized into winter (late October to late March) and summer (the remaining months) schedules. Due to the prevailing wind direction, almost all takeoffs and landings at NBIA are to the east. An Automatic Dependent Surveillance-Broadcast (ADS-B) receiver was installed to collect the flight route information. The estimation was made based on the flight data logged for the whole survey period in one week. The flight data log was obtained from the airport office and compared with the seasonal average traffic to ascertain that the estimated period was representative of the noise situation. The validity of estimated noise levels was confirmed by comparing these noise levels with the measured noise data during the same period. According to the flight logs, the average arrivals and departures at NBIA in a day were counted and classified into day, evening, and night periods as outlined above. These data were then used to calculate the *L*_den_ and *L*_night_ values.

## 3. Results

### 3.1. Demographic Data of the Surveys’ Respondents

A total of 623 and 132 responses were obtained in the first and second follow-up surveys, respectively. Demographic data of the respondents of all the surveys since 2014 was summarized in Table 2. A high response rate was achieved in all the surveys. In recent surveys, the proportions of female respondents are slightly higher than those of males. The respondents aged over 60 years accounted for less than 30% of the total number of respondents in all surveys. These proportions are consistent with Vietnam’s young population structure. There is no significant difference between demographic data in the follow-up surveys and the previous surveys, except the proportion of respondents living in the area for less than five years and the proportion of employed respondents in the second follow-up survey.

### 3.2. Increase in Number of Flights and Noise Levels

The numbers of flights operated and passengers at NBIA have increased significantly over the past five years. Table 3 shows the average number of daily flights operated by NBIA during each survey period. 

Evidently, the number of flights observed in 2018 was about 1.8 times more than that in 2014 and 1.5 times more than that in 2015. Clearly, the number of flights increased sharply after the new terminal building was put into operation and has gradually increased since then. It is worth noting that the most recent number of nighttime flights in the 2018 survey increased six times and four times, compared to September 2014 “before the new terminal building opened” survey and the latest “after the opening” survey in September 2015, respectively. Especially, the recent number of flight events at night has increased sharply and occupied about two-fifths of the total number of flights. The increase of nighttime flights is due to the rapid growth of low-cost carriers which prefer operation at nighttime (22:00–6:00) for a cost-saving benefit. This trend seems to reduce flight frequency in the evening (18:00–22:00) as observed in the 2018 survey.

The noise levels estimated using INM were compared to the noise levels data derived from the field measurement conducted at the corresponding sites in the 2017 survey to verify the validity of the noise estimation. The root means square differences (RMS) between the predicted estimates and the corresponding measured values were 2.4 and 3.9 for *L*_den_ and *L*_night_, respectively. These discrepancies are well–accepted considering that the 2–4 dBA difference in sound level is barely noticeable to the human ear [25]. Regarding estimated *L*_den_ values, the largest deviation was found at Site 5 (−5.1 dB), A12 (+3.9 dB), and A13 (+4.2 dB). When excluding those sites RMS between the predicted estimates and the corresponding measured values of the other sites is 1.3. Regarding estimated *L*_night_ values, the deviation found at these sites are −7.6 dB, +3.7 dB, and 5.9 dB. When excluded estimated *L*_night_ values of those sites RMS is 3.1. The location of the setting point for noise measurement at Site 5 was at the edge of the residential area, right below the center of the aircraft landing track at the closest distance to the airport among other houses at the same site. Meanwhile, because A12 and A13 are located far from the airport to the north without a flight path above and almost unexposed to aircraft noise. These features can cause a significant difference between the actual measurement value at such a special point and the overall calculated value for the area on a noise contour.

Table 4 shows the noise levels obtained during each survey period. In Table 4, the noise levels until 2015 are measured values, and that from 2017 are predicted values. *L*_den_ obtained at the surveyed sites were investigated in all the five surveys ranged from 45 to 66 dB in 2014 and 44 to 73 dB in 2018. Especially, *L*_night_ was found to increase more than 10 dB from 2014 to 2018 at Sites A4 and A5. This result is consistent with the sharp increase in the flight operations during the nighttime at NBIA.

### 3.3. Changes in General Annoyance and Sleep Effects

As shown in Table 5, there is a dramatic increase in % ISM at Site A5, which increased from 17% in September 2015 to 44% in November 2017. This result was consistent with a 2 dB increase in the nighttime noise level measured at Site A5. However, the same trends were not observed in the general annoyance defined by % HA. Despite a slight increase in *L*_den_ between 2015 and 2017, % HA decreased at Sites A7 and A8 which located under the take-off path of aircraft. Among the sites under the landing path, % HA increased remarkably at Site A3 from 65% in 2017 to 96% in the 2017 survey, then decreased to 60% in the 2018 survey. The highest % HA in the survey 2018 was found at the two sites having the highest *L*_den_, Sites A5 (90%) and A8 (80%).

% ISM decreased from 44% in the 2017 survey to 40% ISM in the 2018 survey when *L*_night_ increased 8 dB. Though the sound insulation capacity of the house, the number of the households using an air conditioner and the floor area was not included in the questionnaire item of the 2018 survey. It could be observed during the field investigation that there was a trend of renovating the residences at Site 5. Data in Table 6 shows that there are more houses rated to have good sound insulation, more households using air conditioners, and more houses having floor area more than 100 m^2^. All of these improvements in residential conditions were assumed to continue in the 2018 survey and eased the negative effect on sleep possibly caused by noise exposure and made % ISM to decrease while night noise exposure increased in the 2018 survey.

Logistic regression analysis was applied to establish an exposure–response relationship for each survey. Figure 2 shows a comparison of (a) *L*_den_–% HA and (b) *L*_night_–% ISM relationships established by using data obtained from all the surveys. The *L*_den_–% HA relationships of the follow-up survey in 2017 and 2018, which were conducted about 3 and 4 years after the step change, are lower than those of the 2015 surveys which were carried out 3 and 8 months after the change occurred. The exposure-response relationship established in the follow-up study in 2018 located closer to the relationship established in the survey before the change but significantly higher than that in the European Union Position paper [4]. 

In other words, the change effect due to the step change seems to decline over time but remains higher than that of the steady–state at the same noise levels. However, the L_night_–% ISM relationships obtained through the 2017 survey data were quite high in the level range exceeding 55 dB, while that of the 2018 survey was higher than the relationships obtained in the previous survey in the range below 55 dB. This result indicates that the causal structure of insomnia might be different from that of annoyance. This discrepancy should be explained not only by the amount of noise exposure but also by other various non-acoustical factors.

### 3.4. Effects of Residential Factors and Changes of Noise Exposure on Annoyance and ISM

As seen in the previous section, the annoyance and sleep effects were affected not only by the survey years, but also by changes in exposure levels. Since non-acoustic factors were considered to influence reported aircraft noise annoyance and activity disturbance as significantly as the noise exposure level [16,17], in this section, residential factors such as length of residence, total floor area of the house, evaluation on sound insulation, location of the bedroom, air-conditioner installation are considered to be factors related to respondents’ reactions to noise. In all the surveys, the respondents were asked to evaluate the capacity of sound insulation of their house on a five-point scale from 1 (extremely good) to 5 (extremely bad). A house which was evaluated as having “4. Bad” or “5: Extremely bad” sound insulation was counted as “bad sound insulation”.

The average data of these factors obtained from 13 survey sites are summarized in Table 6. The short length of residence was assumed to increase the respondents’ negative reaction to aircraft noise due to insufficient time to adapt to the living environment near the airport. The respondents living in larger houses with good insulation capacity, a bedroom not facing the road, and an installed air-conditioner were assumed to be less affected by noise than those living in smaller houses, with bad insulation capacity, bedroom facing the road, and no air-conditioner installed. The survey result shows that more air-conditioners were used after 2015. Within more than two years, corresponding to the positive change in the economy, the living amenities of the residents around NBIA has been improved, including the increased use of air conditioners. The percentage of the length of residence that is less than 5 years has decreased in recent surveys.

A multiple logistic regression analysis was conducted to determine the change in the relationships between noise exposure and community response, represented by the correlations between *L*_den_ and the percentage of highly annoyed respondents, and between *L*_night_ and ISM, moderated by the effect of the noise exposure change, residential factors which listed in Table 6, and personal traits such as sex, age, and sensitivity to noise (Table 7 and Table 8). 

The noise exposure change was represented by the difference in noise levels between the after–change surveys and those measured in the first survey (before the change), Δ*L*_den_ and Δ*L*_night_ categories in form of dummy variables. In particular, regarding analysis for annoyance and *L*_den_ association, four dummy variables were created by combining the survey year and the change in the noise levels as follows:Δ*L*_den_ ≤ 0, 2nd & 3rd: Among the respondents that participated in the 2nd and 3rd surveys, the value of 1 was given to those who lived in the area such that *L*_den_ was similar to or lower than that of the 1st survey, and otherwise 0.Δ*L*_den_ > 0, 2nd & 3rd: Among the respondents that participated the 2nd and 3rd surveys, the value of 1 was given to those who lived in the area such that *L*_den_ was higher than that of the 1st survey, and otherwise 0.Δ*L*_den_ ≤ 0, 4th & 5th: Among the respondents that participated in the 4th and 5th surveys, the value of 1 was given to those who lived in the area such that *L*_den_ was similar to or lower than that of the 1st survey, and otherwise 0.Δ*L*_den_ > 0, 4th & 5th: Among the respondents that participated in the 4th and 5th surveys, the value of 1 was given to those who lived in the area such that *L*_den_ was higher than that of the 1st survey, and otherwise 0.

The same categories were created for ISM and *L*_night_. According to the results obtained from logistic regression analysis, the noise change and survey factor represented by the four dummy variable significantly affected the prevalence of annoyance and ISM except in the category of “Δ*L*_den_ ≤ 0, 4th & 5th” in the estimation for annoyance and “Δ*L*_night_ ≤ 0, 2nd & 3rd” and “Δ*L*_night_ ≤ 0, 4th & 5th” in the estimation for ISM. 

Significant associations were found between *L*_den_ and annoyance; and between *L*_night_ and ISM. Personal and residential factors such as noise sensitivity, length of residence, and the evaluation of sound insulation had a significant effect on the prevalence of annoyance. Meanwhile, respondents’ sex, noise sensitivity, and the evaluation of sound insulation had a significant effect on the prevalence of ISM. The prevalence of annoyance and ISM were significantly affected by the noise sensitivity factor. The variable representing the interaction of noise sensitivity and noise exposure, noise sensitivity * *L*_den_ (a) (Table 7) and noise sensitivity * *L*_night_ (Table 8), had a significant effect on annoyance and ISM, respectively. It is worth noting that the coefficient of the interaction between noise level (*L*_den_ or *L*_night_) and noise sensitivity is negative in both models. It indicated that the effect of noise sensitivity decreased when noise exposure increased, and vice versa.

The exposure-response relationships found in Table 7 and Table 8 are presented in the form of graphs in Figure 3. Figure 3a compares the *L*_den_–% HA relationships in the 1st survey and four categories of Δ*L*_den_ and Figure 3b compares the *L*_night_–% ISM relationships in the 1st survey and four categories of Δ*L*_night_ which are listed in Table 7 and Table 8. The categories of noise exposure level which is less than or equal to the first survey are slightly higher and almost coincided with the curve drawn for the first survey in the case of annoyance and insomnia, respectively. On the other hand, the categories of noise level which were greater than those in the 1st survey were significantly higher than those of the curve drawn for the 1st survey.

It is worth noting that the difference from the 1st survey was smaller in the 2017 and 2018 surveys than in the 2015 surveys, regardless of the increase or decrease in Δ*L*_den_. In other words, the excess response regarding noise annoyance in the 2015 survey shortly after the change in noise became more stable in the follow-up surveys. However, the *L*_night_–% ISM relationships established for the category of noise level obtained from the two follow-up surveys, which was greater than those measured in the first survey (“Δ*L*_night_ > 0 in 4th and 5th surveys “category), were higher than those of the 2015 surveys (“Δ*L*night > 0 in 2nd and 3rd surveys” category). In other words, the excess response observed after the change in noise did not decrease over time because the effect of noise on sleep increased. This finding is consistent with the increase in the number of flight movements at night around NBIA. This may negatively affect the quality of sleep and the health of residents living near the airport and result in the community’s excess response to aircraft noise.

### 3.5. Health Effects

In the 2017 survey, blood pressure data were collected through a self-report method. However, because many respondents did not know their blood pressure and thus did not report this data, both self-reported data and data by measurement were collected during the 2018 survey (Table 9). It is worth noting that there was a considerable difference between the reported data and measured data. The correlation coefficient between self-reported blood pressure data and those collected by measurement was 0.378 for systolic blood pressure and 0.393 for diastolic blood pressure. In this study, high blood pressure (HBP) was defined as the systolic and diastolic blood pressure being higher than 140 and 90 mmHg, respectively [26].

A comparison was made to examine the relationship between *L*_den_ and % high blood pressure (Table 10). By running the logistic regression model with data on the possibility of high blood pressure at different noise exposure level ranges, the Wald test was used to determine whether the *L*_den_ could be used to predict or correlate to % high blood pressure (% HBP). The *p*-value shows that *L*_den_ was significantly associated with % HBP in the 2017 survey with self-reported data. This association was not observed in both self–reported and measured data of the 2018 survey.

Multiple logistic regression analysis was applied to the data of the 2018 survey to investigate the relationship between high blood pressure and noise level (Table 11). In this analysis, age, general health status, stress, noise sensitivity, medical issues, drinking, and smoking were applied to the model as influencing factors. The response data of each factor, except the noise level, were categorized into “positive” and “negative.” The “negative” category included responses that possibly cause high blood pressure. Actually, persons who were over the age of 60 years, very or extremely sensitive to noise, negative about their health status, often consumed alcohol, and smoked were categorized as “negative” Particularly, in an open question about the respondents’ medical history, all responses that mentioned diseases that could lead to high blood pressure such as diabetes, thyroid, cardiovascular disease, vestibular disorder, blood lipids, and prostate hypertrophy, were categorized under “negative.” The results shown in Table 11 indicate that “age” and “noise sensitivity” are the most significant factors causing the high blood pressure condition. The variable representing interaction effect of noise exposure and sensitivity was taken into the analysis but found to have no significant effect on the prevalence of high blood pressure. Meanwhile, the association between the level of aircraft noise and high blood pressure was not significant in the 2018 survey.

## 4. Discussion

### 4.1. Declination of Excess Response Over Time

The results of this study suggested that an excess response occurred when Δ*L*_den_ > 0. Under response was observed when Δ*L*_den_ ≤ 0. Meanwhile, regarding sleep effects, excess response was found with Δ*L*_night_ > 0 around NBIA. This result supports the findings of the previous studies by Brink et al. [7], Fidell et al. [8] and Breugelmans et al. [26] which provided evidence showing that the change in response to noise exposure was an excess response to the intervention.

Besides, the curves for the Δ*L*_night_ ≤ 0 in the 2015 surveys and the two follow-up surveys were almost consistent with the curve in a steady–state condition. In other words, no change effect was observed with sleep effects when the noise level remained unchanged or decreased. On the other hand, the excess response in Δ*L*_night_ > 0 was found in the surveys soon after change occurred around NBIA as reported in the study in 2015 [12] and the follow-up surveys. Furthermore, the excess response found in 2017 and 2018 slightly increased but almost similarly to that in 2015 when *L*_night_ increased from the first survey. The findings in NBIA were partly similar to a study conducted at Amsterdam Schiphol Airport by Breugelmans et al. [27], which found that the excess response gradually decreases within two years, in regard to noise annoyance; but differed from the finding that there was no discrete indication of an overreaction in severe sleep disturbance due to the sudden change in noise exposure. This study determined that the effect of change regarding noise annoyance was significant immediately after the completion of the new terminal building of NBIA, and became less significant afterwards, but the change effects on insomnia persisted in sites exposed to higher noise levels. The relationships between aircraft noise exposure and annoyance and effects on sleep were considered in the Environmental Noise Guidelines for the European Region by the World Health Organization (WHO) based on the systematic reviews of evidence from individual studies in which the effect of aircraft noise on self-reported annoyance and sleep outcome were measured [28,29]. The stricter limits stipulated for aircraft noise that were recommended based on these relationships were criticized for validity as they included the results of the surveys conducted at the airports which had undergone the change situation such as after the opening of a new runway or an increase in the number of flights resulting in higher prevalence of annoyance or insomnia [30]. The findings from this study suggest that the excess response due to the change effect may align to the condition before the change occurred, in several years. Criteria should be set for systematic reviews in future. That is, clear criteria for the individual studies on the change effects of aircraft noise to be included in or excluded from the systematic review for future guidelines are necessary.

### 4.2. Effects of Non-Acoustic Factors

Nguyen et al. developed the structural equation model to assess the effects of non-acoustic factors on degrees of road traffic and aircraft noise annoyance [31]. This study found that the aircraft noise annoyance in Hanoi is mainly influenced by sensitivity. In the surveys considered in the present study, noise sensitivity has a significant effect on the evaluation of annoyance, insomnia, and HBP. Among residential factors that are considered to relate to respondents’ reactions to noise in this study, evaluation of sound insulation and length of residence had a significant effect on the prevalence of annoyance, while only evaluation of sound insulation affected the prevalence of insomnia. The sound insulation capacities of more houses were improved, and the percentage of the lengths of residence that were less than 5 years decreased in the follow-up surveys. The changes in these residential factors may ease the negative effects of an increase in noise exposure and cause the excess response to decline. This result is different from the findings of Fields, which stated that there is not a simple adaptation to noise with increasing years of residence [32].

According to the analysis in Section 3.4, in the 2nd and 3rd surveys, the prevalence of annoyance increased even if *L*_den_ decreased, while such a significant increase was not observed in the 4th and 5th surveys and the exposure-response relationships were almost identical to that in the 1st survey. This result suggests that the change effect may relate to not only acoustical factor but other psychological factors, such as cognition and attitude to the expansion of airport operation as discussed in the previous study [13]. However, there was no increase in the prevalence of ISM when *L*_night_ decreased. This suggests that sleep may be less affected by cognition or attitude to the noise source as annoyance. The prevalence of ISM when Δ*L*_night_ > 0 found in both the surveys after the change in 2015 and the follow-up surveys in 2017 and 2018 were almost the similar but higher than that of the 1st survey. In this study, the definition of ISM is based on the frequency of symptoms of sleep effects so that ISM can be considered to be independent of subjective factors. However, symptoms such as trouble with sleep; and being sleepy during daytime and unable to work well more than three times a week were evaluated using the self-report method. Therefore, it was not possible to eliminate the effect of subjective factors. Furthermore, considering the habituation of sleep to the living environment, it is unclear whether we can conclude that there is no excess response in the aspect of sleep effects due to the step change.

In order to further examine the interaction effect on annoyance and ISM of noise sensitivity and noise exposure factors, the *L*_den_–% HA relationship and *L*_night_–% ISM relationship were compared between the sensitive and insensitive respondents (Figure 4). The curves in the model of ISM are reversed at the noise level range of above 63 dB, as a whole, the sensitive group is more responsive to noise exposure than the insensitive group regarding both the annoyance and ISM. This result indicates that the effect of noise sensitivity is reduced in the areas having high levels of noise exposure, or the higher level of noise exposure might saturate the effect of noise sensitivity. In other words, the noise sensitivity was more pronounced at moderate noise level exposed than at high noise level exposed areas.

In the study on the impact of proximity to an airport on health-related quality of life (QOL) of noise-sensitive people by Welch et al., the noise sensitivity has a significant effect on QOL in noise-exposed areas (near the airport), but not so much in non-exposed areas (not near the airport) [33]. In other words, in non-exposed areas, regardless of being sensitive or insensitive, the QOL is not affected. However, in an exposed area, the more sensitive resident will have a more reduced self-reported quality of life. If annoyance and ISM are assumed to relevant to QOL, the finding of the present study is not entirely consistent but uncontradictory with that of Welch et al. ‘s study. In the present study, the effect of noise sensitivity was assessed in a range from low to high noise exposure in the areas around NBIA and found to be saturated by the high noise exposure.

### 4.3. Health Effects of Aircraft Noise

The association between exposure to aircraft noise (*L*_den_) and HBP was identified with self-reported HBP data of the 2017 survey, and unidentified with both self-reported and measured HBP data of the 2018 survey. The systematic review on environmental noise and cardiovascular and metabolic effects by Van Kempen et al. [34], which evaluated 40 studies that investigated the impact of noise from air, road, rail traffic and wind turbines on the risk of hypertension showed the uncertainty on the relation between hypertension and traffic noise exposure. The quality of the evidence was rated as “very low” mainly because the response rate was low and hypertension was assessed by means of self-report only. In the 2018 survey, blood pressure data were obtained through both self-reported and measurement data. However, a low correlation between self-reported and measured HBP data was observed in this study. This result indicates that further efforts are required to define the method of blood pressure data measurement so as to improve the quality of the evidence supporting an association between aircraft noise exposure and hypertension.

### 4.4. Limitations and Implications

The present study had a few limitations. First, our surveys were cross-sectional studies, as opposed to cohort studies which are preferable for this kind of research. We recently initiated a cohort survey on health effects in Ho Chi Minh City. Our next investigation based on cohort surveys will further verify the relationships between aircraft noise exposure and its effects on sleep and health among the populations living near the airports that have undergone change. Second, all of the data on public reactions was derived from face-to-face interviews conducted in the order of father, mother, and adults other than parents in each house. Though this method seems to be almost similar to random sampling and the demographic distribution almost matches the Vietnamese census, it would have been better if the random sampling method was applied. However, a random sampling based on residence registration may not be possible at present in Vietnam. Third, the limited amount and uncertainty of the health data prevented us from affirmatively concluding the association between noise exposure and health indicators. In future study, we will work with the local hospitals and clinics to access the respondents’ health data. Fourth, noise exposure data of the 2014–2015 surveys were obtained from field measurements, while those of the 2017–2018 survey were estimated from the noise map. Though the predicted and measured values were almost consistent except for some sites with special conditions, obtaining noise exposure data from the noise maps throughout the entire study would have been more favorable. Despite the above limitations, the contributions of the present study are relevant to the policymakers, the aviation authority, and the environmental managers. The results of this study provide scientific evidence regarding the association of aircraft noise change and the community health and thus inform policies and guidelines aimed at protecting and improving community health for the population living in the vicinity of airports. This study also suggests a practical methodology for assessing the effects of sound environment change caused by traffic development for developing countries.

## 5. Conclusions

In this study, the responses obtained in the follow-up surveys are higher than those obtained before the opening of the new terminal at the end of 2014 under the same noise level. However, it seems that the change effect due to the operational change is observed to decrease in the follow-up study with regard to annoyance and remains the same with regard to insomnia. The results show that the aircraft noise level *L*_den_ was significantly associated with self-reported % HBP data in the 2017 survey; however, it was not correlated with the self-reported and measured data of the 2018 survey. Personal characteristics such as age and noise sensitivity are the most important moderators of the high blood pressure condition. Noise sensitivity is also a significant factor that affected the prevalence of annoyance and insomnia among the residents living around NBIA. It was found that the effect of noise sensitivity might be saturated with the respondents living in the areas having high levels of noise exposure.

A higher prevalence of insomnia was observed during the survey period when the night flight operation was enhanced. An increase in the number of flights operated at night negatively affected the quality of sleep. Protecting the living environment in the vicinities of the airports in Vietnam should be considered by improving the residence quality, restricting night operation, and formulating policy regarding aircraft noise. In further studies on the health impact of noise on residents living around the major airports in Vietnam, an appropriate method of collecting health data should be considered.

## Figures and Tables

**Figure 1 ijerph-17-02597-f001:**
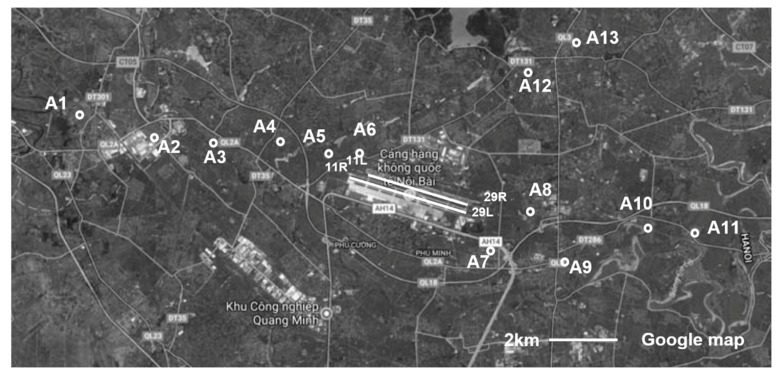
Map of all surveyed sites.

**Figure 2 ijerph-17-02597-f002:**
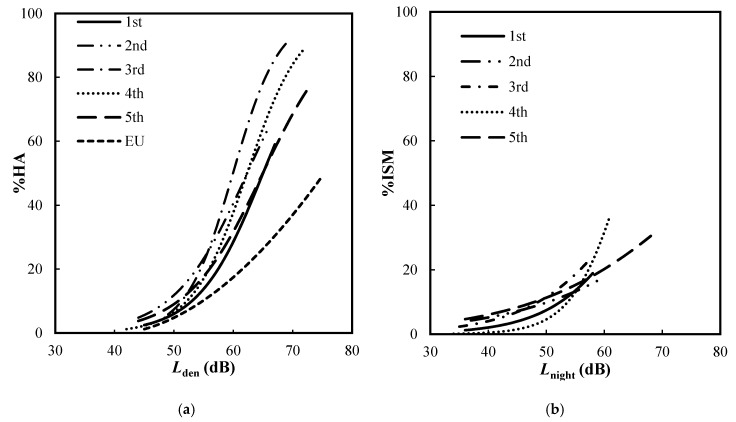
Comparison of the relationships synthesized from the data of each survey from 2014 to 2018. (**a**) Lden–% HA relationships; (**b**) Lnight–% ISM relationships.

**Figure 3 ijerph-17-02597-f003:**
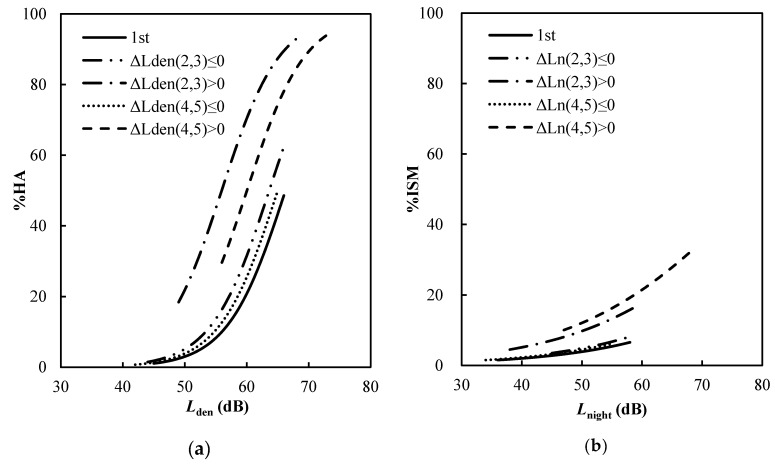
Comparison of (**a**) Lden–% HA and (**b**) Lnigh–% ISM relationships adjusted by modifying factors in the 1st survey (reference curve) and sites classified with ΔLden and ΔLnight; non-acoustic variables including sex, age, noise sensitivity, length of residence, floor area and sound insulation; and interaction of noise sensitivity and Lden (**a**) and Lnight (**b**). The numbers in the parentheses show the survey numbers. For example, “ΔLden ((2,3) ≤ 0)” means “ΔLden (in the 2nd and 3rd surveys) ≤ 0”.

**Figure 4 ijerph-17-02597-f004:**
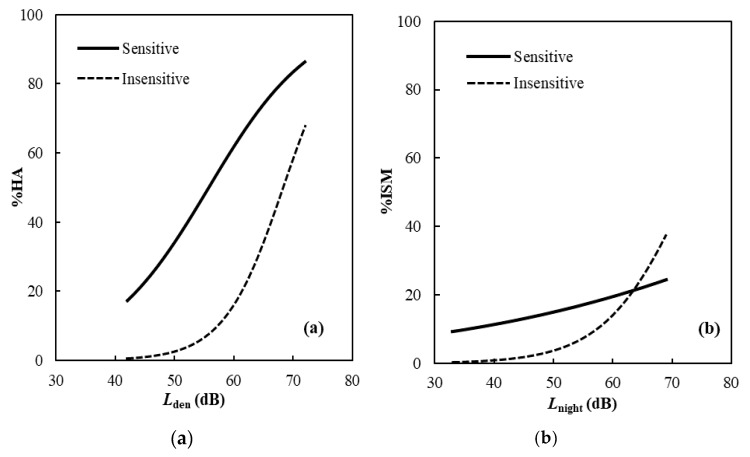
Comparison of (**a**) Lden–% HA and (**b**) Lnigh–% ISM relationships between the noise sensitive and insensitive respondents in all the surveys.

**Table 1 ijerph-17-02597-t001:** Questions used to define insomnia.

18. Please Answer This Question Concerning Your Sleep:			
**(1) Do you have any trouble with your sleep?**			
No ( )			
Yes ( )			
**If you answered “Yes” to the above question, please choose appropriate numbers for each item.**			
	1 Occasionally	2 Once or twice a week	3 More than 3 times a week
(1) Difficult to fall asleep	( )	( )	( )
(2) When awakened during the night, it is difficult to sleep again.	( )	( )	( )
(3) Awakened early in the morning	( )	( )	( )
(4) Do not feel as having slept well the next morning.	( )	( )	( )
(5) Sleepy during daytime and cannot work well	( )	( )	( )
(6) Others ( )	( )	( )	( )
**(2) If you have trouble with your sleep, do you think that it is due to the aircraft noise?**			
No ( )			
Yes ( )			

**Table 2 ijerph-17-02597-t002:** Demographic data of the respondents in all the surveys.

Items		Surveys					Vietnamese Census (2018)
		Sep 2014	Mar 2015	Sep 2015	Nov 2017	Aug 2018	
Number of respondents		890	1109	1286	623	132	
Response rate (%)		68.5	85.3	98.8	95.8	83.3	
Gender	Male	54.1	52.4	49.4	47.7	40.9	49.5
	Female	45.9	47.6	50.6	52.3	59.1	50.5
Age	20–50 years	82.2	84.3	84.7	75.5	71.2	88.6
	≥60 years	17.8	15.7	15.3	24.5	28.8	11.4
Length of residence	Under 5 years	15.5	10.1	10.7	9.0	6.4	
	5 years or more	84.5	89.9	89.3	91.0	93.6	
Occupation	Employment	53.5	60.3	60.4	51.4	75.0	56.5
	Student, housewife, retired, unemployed	46.5	39.7	39.6	48.6	25.0	43.5

**Table 3 ijerph-17-02597-t003:** Average number of aircraft noise events.

Time Period		Surveys				
	Operation Modes	Sep 2014	Mar 2015	Sep 2015	Nov 2017	Aug 2018
Day (6:00–18:00)	Arrival	84	104	100	120	141
	Departure	90	109	107	135	123
	Total	174	213	207	255	264
Evening (18:00–22:00)	Arrival	32	43	39	47	12
	Departure	16	27	22	35	13
	Total	48	70	61	82	25
Night (22:00–6:00)	Arrival	9	16	14	38	77
	Departure	21	26	25	36	94
	Total	30	42	39	74	171
All day	Arrival	125	163	153	205	230
	Departure	127	162	154	206	230
	Total	252	325	307	411	460

**Table 4 ijerph-17-02597-t004:** *L*_den_^a^, *L*_night_^b^, and their changes from the 1st to 2nd, 3rd, 4th or 5th surveys.

Site	*L* _den_					*L* _night_					*∆L* _den_				*∆L* _night_			
	Sep 2014	Mar 2015	Sep 2015	Nov 2017	Aug 2018	Sep 2014	Mar 2015	Sep 2015	Nov 2017	Aug 2018	Mar 2015	Sep 2015	Nov 2017	Aug 2018	Mar 2015	Sep 2015	Nov 2017	Aug 2018
A1	55	55	53	53	55	45	46	45	44	48	0	−2	−2	0	1	0	−1	3
A2	55	56	54	56	58	45	48	46	47	51	1	−1	1	3	3	1	2	6
A3	62	64	62	60	62	53	56	55	51	56	2	0	−2	0	3	2	−2	3
A4	54	56	57	61	63	46	48	48	52	56	2	3	7	9	2	2	6	10
A5	61	61	68	71	73	51	53	59	61	69	0	7	10	12	2	8	10	18
A6	65	64	64	64	65	50	57	56	56	58	−1	−1	−1	0	7	6	6	8
A7	66	62	62	64	67	55	56	55	54	60	−4	−4	−2	1	1	0	−1	5
A8	66	66	65	65	67	58	58	58	55	60	0	−1	−1	1	0	0	−3	2
A9	63	60	63	65	66	55	53	56	56	60	−3	0	2	3	−2	1	1	5
A10	60	58	59	58	60	52	52	53	48	53	−2	−1	−2	0	0	1	−4	1
A11	60	57	59	57	59	52	50	52	48	52	−3	−1	−3	−1	−2	0	−4	0
A12	45	45	49	42	44	36	38	39	34	36	0	4	−3	−1	2	3	−2	0
A13	47	44	51	42	44	36	38	44	34	36	−3	4	−5	−3	2	8	−2	0

^a^ Day-evening-night-weighted sound pressure level; ^b^ Nighttime equivalent continuous sound pressure level.

**Table 5 ijerph-17-02597-t005:** Percentage of highly annoyed (% HA) and percentage of insomnia (% ISM).

Site	% HA					% ISM				
	Sep 2014	Mar 2015	Sep 2015	Nov 2017	Aug 2018	Sep 2014	Mar 2015	Sep 2015	Nov 2017	Aug 2018
A1	8	6	2	0	20	1	1	0	0	20
A2	9	36	29	14	20	0	7	3	4	20
A3	59	71	65	96	60	17	20	22	2	22
A4	48	83	92	78	60	18	27	22	19	20
A5	48	74	96	92	90	9	34	17	44	40
A6	71	64	84	83	60	5	8	20	17	10
A7	44	12	61	10	20	5	18	9	0	10
A8	58	55	69	33	80	33	1	7	8	10
A9	28	38	56	53	10	7	6	24	11	10
A10	10	10	28	34	0	6	5	12	10	10
A11	9	6	11	12	40	0	4	5	0	30
A12	0	0	2	0	9	0	0	1	0	0
A13	0	0	3	0	0	6	1	1	0	0

**Table 6 ijerph-17-02597-t006:** Changes in residential factor through the surveys from 2014 to 2018.

Residential Factors (%)	Sep 2014	Mar 2015	Sep 2015	Nov 2017	Aug 2018
Length of residence ≤ 5 years	15.5	10.1	10.7	9.0	6.4
Floor area ≤ 100 m^2^	40.6	71.4	67.7	51.1	-
Bad sound insulation	33.0	31.0	38.9	32.4	-
Bedroom facing road	-	35.3	31.0	44.2	-
No air-conditioner installed	-	71.5	71.2	50.1	-

**Table 7 ijerph-17-02597-t007:** Multiple logistic regression for annoyance (Generalized R^2^: 0.3632; AUC (Area under the curve): 0.8787).

Item	Category	Estimate	Std Error	*p*-Value	Odds Ratio	Lower 95%	Upper 95%
Annoyance							
Intercept		−18.008	1.227	<0001			
*L* _den_ ^a^		0.260	0.020	<0001	1.297	1.248	1.349
Δ*L*_den_ ^b^	1st Survey				1		
	Δ*L*_den_ ≤ 0, 2nd & 3rd	0.563	0.140	0.0001	1.757	1.335	2.311
	Δ*L*_den_ > 0, 2nd & 3rd	2.206	0.191	<0001	9.079	6.245	13.199
	Δ*L*_den_ ≤ 0, 4th & 5th	0.267	0.226	0.2362	1.307	0.839	2.034
	Δ*L*_den_ > 0, 4th & 5th	1.331	0.306	<0001	3.785	2.080	6.888
Sex	Male				1		
	Female	0.153	0.100	0.1283	1.165	0.957	1.418
Age	≤ 60 years				1		
	> 60 years	−0.055	0.137	0.6887	0.947	0.724	1.238
Noise sensitivity	Not sensitive				1		
	Sensitive	2.065	0.119	<0001	7.883	6.244	9.951
Noise sensitivity * *L*_den_		−0.096	0.025	0.0001			
Length of residence	>5 years				1		
	≤ 5years	−0.446	0.169	0.0083	0.640	0.460	0.891
Floor area	> 100 m^2^				1		
	≤ 100 m^2^	−0.044	0.108	0.6831	0.957	0.774	1.183
Sound insulation	Good				1		
	Not good	0.367	0.104	0.0004	1.443	1.176	1.770

* Odds ratio in 1 dB change. ^a^ Day-evening-night-weighted sound pressure level. ^b^ The difference in noise levels between the after–change surveys and those measured in the first survey.

**Table 8 ijerph-17-02597-t008:** Multiple logistic regression for insomnia (Generalized R^2^: 0.1516; AUC: 0.747).

Item	Category	Estimate	Std Error	*p*-Value	Odds Ratio	Lower 95%	Upper 95%
Intercept		−11.176	1.388	<0001			
*L* _night_ ^a^		0.140	0.026	<0001	1.150	1.211	0.869
Δ*L*_night_ ^b^	1st Survey				1		
	Δ*L*_night_ ≤ 0, 2nd & 3rd	0.239	0.334	0.4757	1.269	0.659	2.445
	Δ*L*_night_ >0, 2nd & 3rd	0.984	0.223	<0001	2.676	1.728	4.144
	Δ*L*_night_ ≤ 0, 4th & 5th	0.149	0.425	0.7261	1.161	0.504	2.672
	Δ*L*_night_ > 0, 4th & 5th	1.223	0.302	0.0001	3.398	1.881	6.139
Sex	Male				1		
	Female	0.427	0.129	0.0010	1.533	1.190	1.974
Age	≤ 60 years				1		
	> 60 years	0.142	0.167	0.3930	1.153	0.832	1.599
Noise sensitivity	Not sensitive				1		
	Sensitive	1.405	0.172	<0001	4.077	2.911	5.712
Noise sensitivity * *L*_night_		−0.147	0.031	<0001			
Length of residence	> 5 years				1		
	≤ 5 years	−0.181	0.233	0.4364	0.834	0.529	1.317
Floor area	>100 m^2^				1		
	≤100 m^2^	−0.145	0.139	0.2955	0.865	0.659	1.135
Sound insulation	Good				1		
	Not good	0.362	0.132	0.0061	1.437	1.109	1.861

* Odds ratio in 1 dB change. ^a^ Nighttime equivalent continuous sound pressure level. ^b^ The difference in noise levels between the after–change surveys and those measured in the first survey.

**Table 9 ijerph-17-02597-t009:** Blood pressure data including self-reported and measured data at all sites of the follow-up surveys.

Site	Nov 2017 (Self-Reported)		Aug 2018 (Self-Reported)		Aug 2018 (Measured)	
	No. of Response	% HBP ^a^	No. of Responses	% HBP	No. of Response	% HBP
A1	46/50	37	-	-	10/10	50
A2	12/50	67	1/10	100	10/10	30
A3	17/50	0	3/10	33	10/10	10
A4	2/50	50	2/10	50	10/10	20
A5	26/50	27	1/10	0	10/10	0
A6	-	-	1/10	100	10/10	0
A7	13/50	15	1/10	0	10/10	20
A8	4/50	25	-	-	9/10	33
A9	5/47	20	2/10	0	10/10	60
A10	18/50	11	1/10	0	10/10	40
A11	5/50	100	1/10	0	9/10	22
A12	23/50	17	3/10	33	10/11	70
A13	18/50	22	3/10	33	10/11	20

^a^ High blood pressure.

**Table 10 ijerph-17-02597-t010:** Comparison of high blood pressure ratios at different noise level ranges of the 2017–2018 surveys.

		Noise Level Ranges L_den_ ^a^ (dB)					*p*-Value
		<55	55–60	60–65	65–70	> 70	
Nov 2017	% HBP	46	51.4	37.8	100	100	< 0.01
(self-reported)	Response number	40/87	18/35	14/37	4/4	26/26	
Aug 2018	% HBP	50	50	16.7	20	0	0.174
(self-reported)	Response number	3/6	1/2	1/6	1/5	0/1	
Aug 2018 (measured)	% HBP	45	33.3	30	38	30	0.556
	Response number	9/20	10/30	6/20	19/50	3/10	

^a^ Day-evening-night-weighted sound pressure level.

**Table 11 ijerph-17-02597-t011:** The results of multiple logistic regression analysis for investigating the relationship between noise level and high blood pressure.

Item		Estimate	Std Error	*p*-Value	Odds Ratio	Lower 95% CI	Upper 95% CI
Intercept		1.091	2.313	0.6372			
*L* _den_ ^a^		−0.063	0.036	0.0835	0.939	0.870	1.006
Age	20–50 years				1		
	≥60 years	2.422	0.719	<0.001	11.270	2.754	46.118
Self-evaluated Health status	Positive				1		
	Negative	−0.249	0.591	0.6736	0.674	0.245	2.484
Noise sensitivity	Not sensitive				1		
	Sensitive	−2.122	0.934	<0.05	0.128	0.020	0.838
Stress	Positive				1		
	Negative	−1.736	1.266	0.1704	0.120	0.015	2.108
Medical problems	Positive				1		
	Negative	−0.423	0.618	0.4930	0.655	0.195	2.197
Smoking	Non–smoking				1		
	Smoking	0.641	0.696	0.3567	1.899	0.486	7.423
Drinking alcohol	Not drinking				1		
	Drinking	0.762	0.677	0.2603	2.143	0.537	7.758

^a^ Day-evening-night-weighted sound pressure level.

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
