# Peer review of "Effects of Changes in Acoustic and Non-Acoustic Factors on Public Health and Reactions: Follow-Up Surveys in the Vicinity of the Hanoi Noi Bai International Airport"

_ijerph, 2020, doi:10.3390/ijerph17072597_

Round 1

Reviewer 1 Report

General Comments

The manuscript analyses the effects of changes in Acoustic and Non–acoustic factors on public health by five surveys around the residential area of Hanoi Noi Bai International Airport.

Generally, the contents and conclusions of the paper are of particular interest. The title is a brief and accurate indication of the manuscript and the abstract is a good summary of the whole paper. The paper is well organised and the contents are technically accurate. However, some extra information is expected to better understand how “noise estimation” was made on the different surveys (see specific comments). Regarding that, some technical decisions could affect manuscript’s conclusions.

The references are adequate for the contents, complete and all for them are cited in the text.

Specific comments

In section 2.3, it should be properly justify why in some surveys sound pressure level were estimated from field measurements and in other SPL were estimated using the "Interested Noise Model". Important deviations between surveys could be justified by the different “estimation” methodologies used. Regarding field measurements, some extra information should be added (time ponderation, façade distance,…)

In section 3.2, lines 171-191, some deviation in the sites are given, but it is not clear how they were obtained (no table with SPL estimated). Why is said that “these discrepancies are well-accepted”? could you add any reference to justify this assumption? How do you justify deviation of -7.6dB? Table 4 is not clear at all. Not evidences of where this results are coming from. Why all SPL are integer numbers?

In section 3.3, some comments of results of table 5 are made (lines 2,3,4), referring to the “dramatic increase” in site A5 from 2015 to 2017, but why does %ISM reduce from 2017 to 2018, when Ln increases 8 dB? How is figure 2 obtained? “comparison of (a) Lden–%HA and (b) Lnight– %ISM relationships established by using data obtained from all the surveys” but how? Why are survey dates changed by 1st, 2nd,…? In table 6, how were “bad sound insulation” estimated?

Conclusion

Overall, this work is potentially of interest, but some important changes and extra information, that can change some of the conclusions of the manuscript, should be added. I hope my comments will help you.

Reviewer 2 Report

This was a very interesting study consisting of a series of measures of sound levels and annoyance and health data at sites around a Vietnamese airport, allowing the authors to capture change in sound levels as a predictor variable. Analyses revealed similarities and differences from previous research.

The work was generally clearly presented, analyses seemed sensible, and the writing was clear and comprehensible. Limitations were addressed and I believe that the authors’ comments around them were acceptable. I have two very minor points and a slightly more complex recommendation to make to the authors.

Method: How was noise-sensitivity measured?

Table 4: ‘Air-conditioner uninstalled’ sounds as though it is about the removal of existing air-conditioners. From the text, I think ‘No air-conditioner installed’ would be better.

Tables 7, 8, and 11 and supporting text: I don’t think it is right to consider noise sensitivity as a confounding factor. Previous research near airports (e.g. Welch, D., Dirks, K. N., Shepherd, D., & McBride, D. (2018). Health-related quality of life is impacted by proximity to an airport in noise-sensitive people. Noise & Health, 20(96), 171-177) has shown that there is an interaction between noise exposure and noise sensitivity for annoyance/health outcomes wherein effects of noise exposure are reduced or absent in those who are not noise sensitive while being much stronger in those who are. It would be useful and interesting to explore the data here for evidence of an interaction between noise sensitivity and exposures – please could the authors consider adding this to the analyses?
